# Does denitrification occur within porous carbonate sand grains?

Perran L.M. Cook[1], Adam J. Kessler[1], Bradley D. Eyre[2]

[1]Water Studies Centre, School of Chemistry, Monash University, Clayton, Australia

[2]Centre for Coastal Biogeochemistry, Southern Cross University, Lismore, Australia

10      Correspondence to Perran Cook (perran.cook@monash.edu)

**Abstract.** Permeable carbonate sands form a major habitat type on coral reefs and play a major role in organic matter recycling. Nitrogen cycling within these sediments is likely to play a major role in coral reef productivity, yet remains poorly studied. Here, we used flow-through reactors and stirred reactors to quantify potential rates of denitrification and the dependence of denitrification on oxygen concentrations in permeable carbonate sands at three sites on Heron Island, Australia. Our results showed that potential rates of denitrification fell within the range of 2-28 $\mu$mol $L^{-1}$ sediment $h^{-1}$, and were very low compared to oxygen consumption rates, consistent with previous studies of silicate sands. Denitrification was observed to commence at porewater oxygen concentrations as high as 50 $\mu$M in stirred reactor experiments on the coarse sediment fraction (2-10 mm), and at oxygen concentrations of 10- 20 $\mu$M in flow through and stirred reactor experiments at a site with a median sediment grain size of 0.9 mm. No denitrification was detected in sediments under oxic conditions from another sited with finer sediment (median grain size 0.7 mm). We interpret these results as confirmation that denitrification may occur within anoxic microniches present within porous carbonate sand grains. The occurrence of such microniches has the potential to enhance denitrification rates within carbonate sediments, however further work is required to elucidate the extent and ecological significance of this effect.

## 1. Introduction

Nitrogen is typically regarded as one of the key nutrients limiting production in the coastal environment (Howarth and Marino, 2006). Coral reefs are examples of highly oligotrophic environments that are coming under increasing threat from increased nutrient loads (De'ath et al., 2012). Denitrification is a key remedial process in the nitrogen cycle, leading to the conversion of bioavailable nitrate into relatively non-bioavailable nitrogen gas. One of the dominant habitat types on coral reefs are carbonate sands formed from the breakdown of carbonate produced by calcifying organisms (Eyre et al., 2014). Quantifying denitrification in sandy sediments is complicated by the fact that these sediments are permeable, allowing water movement through the sediment, which can both enhance and reduce denitrification (Cook et al., 2006; Kessler et al., 2012; Sokoll et al., 2016). Reproducing these conditions whilst measuring denitrification is difficult and thus, models combined with a mechanistic understanding of the primary controls on denitrification offer a promising approach to quantifying denitrification in these environments (Evrard et al., 2013; Kessler et al., 2012).

For denitrification to take place under anoxic conditions, a supply of nitrate and organic matter, are required. Advective flushing of permeable sediments leads to deeper oxygen penetration into these sediments, which will inhibit denitrification (Evrard et al., 2013; Precht et al., 2004). Nitrate may be supplied to the denitrification zone from either the water column or nitrification within the sediment. It has been reported that nitrification within the sediment may be a significant source of nitrate to fuel denitrificication (Marchant et al., 2016; Rao et al., 2007), although modeling studies suggest that even in the presence of nitrification, flow fields in permeable sediments may lead to little coupling between nitrification and denitrification (Kessler et al., 2013). In systems with high-bottom water nitrate concentrations, high rates of denitrification have been reported (Gao et al., 2012; Sokoll et al., 2016). Previous studies have also reported high rates of denitrification in carbonate sands (Eyre et al., 2008; Eyre et al., 2013b), which is surprising given the low nitrate concentrations in the overlying water and their highly oligotrophic nature. One possible explanation for high rates of denitrification is the coupling of nitrification and denitrification within microniches associated with sand grains (Jahnke, 1985; Rao et al., 2007; Santos et al., 2012). In this hypothesis, mineralization of organic matter within or on sand grains consumes oxygen at a rate greater than it can diffuse into the grain, causing anoxia within the grain. This allows the bulk sediment to be simultaneously oxic, which promotes nitrification, while anoxic sites within grains allow denitrification to take place closely coupled to nitrification. A recent study has shown that incorporating intra-granular porosity into model simulations of column

experiments can lead to better agreement with observations (Kessler et al., 2014) in carbonate sediments, but direct experimental evidence for this phenomenon is lacking.

Direct measurement of oxygen concentrations and denitrification rates within sediment grains is not possible with current technologies. Nevertheless, there are means by which the hypothesis that denitrification takes place within sediment grains can be tested. If denitrification is taking place within anoxic intra-granular niches, then it should be observed under oxygen concentrations above zero within flow through reactors and/or stirred reactors. Another factor complicating our understanding of denitrification are the kinetics of this process relative to total sediment respiration. It has previously been shown that denitrification rates in permeable sediments are very low compared to total respiration (Bourke et al., 2017; Evrard et al., 2013) and that this pattern is consistent globally (Marchant et al., 2016) in silicate sands. There have, however, been no analogous studies on potential denitrification rates relative to respiration rates in anoxic flow through reactors in carbonate sediments.

To address these knowledge gaps, this study had two objectives. The first was to measure potential denitrification rates in carbonate sediments and compare these to total metabolism measured in flow through column experiments. The second was to investigate the effect of oxygen concentration on denitrification using both flow through and stirred reactors to experimentally test the plausibility of anoxic micro niches leading to enhanced rates of denitrification in carbonate sediments.

## 2. Methods

### 2.1 Flow through reactor experiments

Sediments were collected from three sites at Heron Island, 23° 27'S, 151° 55'E in the Southern Great Barrier Reef, Australia October 17-21 2015. Previously reported water column nutrient concentrations at this site are $NO_3^-$ 0.05 – 0.7 $\mu$M, $NH_4^+$ 0.05 – 1.8 $\mu$M, orthophosphate 0.35 – 0.5 $\mu$M and sediment organic carbon content is < 0.24% and benthic chlorophyll $a$ has been reported as ranging from 11 – 15 mg m$^{-2}$ (Eyre et al., 2008; Glud et al., 2008). Site 1 is located adjacent to the Heron Island research station (23° 26' 37" S, 151° 54' 46" E, water depth 0.5 m at low tide), site 2 is located at Shark Bay (23° 26' 37" S, 151° 55' 09" E water depth 0.5 m low tide), and Site 3 is located in the lagoon approximately 4 km east of the island (23° 27' 04" S, 151° 57' 28" E, water depth 2 m at low tide). Site 3 was the most uniformly coarse and permeable site, and site 2 had the smallest median grain size and lowest permeability (Table 1). Sediments were packed into three replicate flow-through reactors (FTRs, 4.6 cm diameter, 4 cm length) for each experiment, as described by Evrard et al. (2013), within 2 hours of collection. Fresh unfiltered seawater was collected from in front of the research station and the columns were percolated at a flow rate of ~200 mL h$^{-1}$. The volume of the FTRs was 66 mL (~33 mL porosity corrected), giving a retention time of ~10 mins (corrected for sediment porosity). Reaction rates were calculated per volume of wet sediment. The flow velocity was 24 cm h$^{-1}$ and was chosen as it was estimated to give a small but easily detectable change in $^{15}$N-$N_2$. This is the upper end of those expected around ripples of 0.14-26 cm h$^{-1}$ (Precht et al., 2004), and higher than those used by Santos et al. (2012). We deliberately did this to ensure any boundary layers at the grain surface were at a minimum and any effect observed here could be ascribed to intra-granular porosity. Diffusive chemical gradients were manipulated by changing oxygen and nitrate concentrations within the flow-through reactors.

Two experiments were undertaken as follows. First, the effect of $NO_3^-$ concentration on denitrification was measured as described by Evrard et al. (2013). Repacked FTRs were percolated with anoxic seawater (30 mins purging with Ar), which was sequentially amended with 18, 37, 75, 150 and 300 $\mu$M $^{15}NO_3^-$. The oxygen concentration at the column inlets and

outlets were monitored in real time using Firesting optical dissolved oxygen flow-through cells (Pyroscience) which had a detection limit of ~3 μM $O_2$ and a precision of ~1%. After ~3 retention times (~30 mins) at each nitrate concentration, a sample of the column effluent was collected directly into glass syringes, and transferred into an exetainer and preserved with 250 μL 50% w/v $ZnCl_2$. Second, the effect of oxygen on denitrification was measured as described by Evrard et al. (2013). Columns were percolated with unfiltered seawater amended with 150 μM $^{15}NO_3^-$ (99% Cambridge scientific), and the oxygen concentration was reduced incrementally in the reservoir by sporadic purging with Ar. Samples of column effluent were collected and preserved as described above.

**2.2 Stirred reactor experiments**

Samples for stirred reactor (SR) experiments were collected from Sites 1 and 2 on March 14, 2017 and sent by overnight courier to Monash University, where they were submerged in aerated artificial seawater made from 'Redcoral' nitrate and phosphate free sea salts at 23°C amended with ~50 μM $^{15}NO_3^-$. SR experiments were undertaken on sieved sediments (<2 mm) at sites 1 and 2 and the coarse fraction (2-10mm) from site 1 in 115 mL glass vessels, capped with a rubber bung ensuring no air bubbles were present as described by (Gao et al., 2010). The reactor was stirred using a magnetic stirrer bar at ~150 rpm which was the speed required to re-suspend all the added sediment (15 mL). Water samples were withdrawn though a port into a syringe over a period of 3-5 hours and simultaneously replaced with the same volume of artificial seawater (~15mL) and preserved for nitrogen isotope analysis as described above. Oxygen was logged using a Firesting (Pyroscience) needle $O_2$ sensor inserted through the rubber bung.

**2.3 Analytical methods**

Samples for $^{28}N_2$, $^{29}N_2$ and $^{30}N_2$ analysis had a 4 ml He headspace inserted into the exetainer, and were shaken for 5 minutes before the headspace gas was analysed on a Sercon 20-22 isotope ratio mass spectrometer, coupled to an autosampler and GC column to separate $O_2$ and $N_2$. Air was used as the calibration standard, and tests showed no false mass 30 signal compared to pure $N_2$ injections. The precision of the analysis of the ratios $^{29}N_2$ /$^{28}N_2$ and $^{30}N_2$ /$^{28}N_2$ was 0.2 and 5 %, respectively. For the analysis of 2 μmol N, this equates to an excess $^{15}N$ of $2.5 \times 10^{-5}$ μmol for $^{29}N_2$ and $7.85 \times 10^{-5}$ μmol for $^{30}N_2$. Assuming all of the $N_2$ production was in the form of $^{30}N_2$, this results in an equivalent detectable production rate of 0.014 nmol mL$^{-1}$ h$^{-1}$ in the column experiments. Rates of denitrification were calculated using the isotope pairing equations (Nielsen, 1992) and we present the total rate of denitrification ($D_{14}+D_{15}$) here. Rates of anammox were estimated based on equation 23 given in Risgaard Petersen et al. (2003). Sediment permeability was measured using the constant head method (Reynolds, 2008) and sediment porosity was measured by drying a known volume of sediment saturated with fresh water. Images of the grains were taken using a Motic dissecting microscope with a 5MP Moticam. The porosity of the sand grains was measured using mercury porosimetry at Particle & Surface Sciences Pty Ltd. Sediment grain size was measured using test sieves with mesh sizes of 2, 1.18, 0.5 and 0.125 mm.

**3. Results**

Sites 1 and 3 had the coarsest median grain size of 0.9 mm, whereas site 2 had a median grain size of 0.7 mm. All sites had 20-30 % sediment with grain sizes in the range of 1.18 – 3mm, the sediment permeability was also similar at all sites ranging from 24-30 $\times 10^{-12}$ m$^2$ and the bulk porosity was similar at all sites ranging from 0.48 – 0.56 at the three sites (Table 1). Images of the sand grains showed them to be porous (Figure 1), and this was confirmed by mercury porosimetry, which revealed the sand grains had a porosity of ~0.32 at sites 1 and 3, site 2 not measured (Table 1).

In the flow through reactor (FTR) experiments rates of denitrification were constant above $NO_3^-$ concentrations of 18 μM at all three study sites, and were highest at site 3 which had the highest sediment oxygen consumption rates and lowest at site 2 which had the lowest oxygen consumption rates (Figure 2). Rates of anammox comprised <16% of nitrogen production (data not shown) in the anoxic flow through columns. Plots of oxygen concentrations showed that concentrations of oxygen at the column inlets dropped in a stepwise manner when they were purged with Ar, and this was reflected at the column outlets with a delay of ~10 minutes consistent with the theoretical column retention time (Figure 3). Rates of denitrification were generally negligible at oxygen concentrations > 0 μM at the column outlets, except for site 1 where denitrification was observed at ~10 μM $O_2$ at the column outlet (Figure 4).

The stirred reactor (SR) experiments showed small amounts of $^{15}N-N_2$ production once the oxygen concentration in the reactor dropped below 20 μM at site 1 (Figure 5). At site 2, $^{15}N-N_2$ production was only observed once the oxygen concentration approached 0 μM. The coarse fraction at site 1 showed mixed results with 2 of the 4 samples showing no clear $^{15}N-N_2$ production above 20 μM $O_2$ and 2 samples showing clear evidence of $^{15}N-N_2$ production at <50 μM $O_2$.

## 4. Discussion

### 4.1 Methodological considerations

Before discussing the results in detail, we briefly consider the methods used here and potential shortcomings. First, we only used one, relatively fast flow rate in these FTR experiments. We chose this flow rate as we estimated this was the maximum flow rate we could use that would minimize boundary layers within the column, while giving detectable production of $^{15}N-N_2$. These flow rates are in the upper range of those previously reported in sediments where porewater flow is driven by flow-topography interactions (Precht and Huettel, 2004), as we expected to be the case here. Second, it is possible that $^{15}N-N_2$ production had not reached a steady state after the manipulation of oxygen and nitrate concentrations within the FTRs. Conceptually, nitrate from the bulk porewater will diffuse into the sediment grain, where denitrification will take place, and the produced $^{15}N-N_2$ will diffuse out again before being washed out of the column. If we use a grain size of 2 mm (twice the median grain size), this means a maximum diffusion distance of ~1 mm to a putative denitrification zone within a sediment grain. The diffusion timescale for nitrate molecule can be calculated using equation 1

$$t = L^2/2D_s \tag{1}$$

where t= time, L= distances and $D_s$ is the diffusion coefficient (Schulz and Zabel, 2005). For nitrate, with a diffusion coefficient of $1.7 \times 10^{-5}$ $cm^2$ $s^{-1}$ at 25°C and a salinity of 35 corrected for a grain porosity of 0.3 according to (Iversen and Jørgensen, 1993) gives a timescale of ~10 mins. For nitrate to diffuse in and $N_2$ to diffuse out, we would therefore expect this to take a maximum of ~20 mins which is less than the time we waited before sampling in the column and the time interval between samples in the SRs. For the coarse fraction used in the stirred reactors, the samples taken were unlikely to represent steady state, and therefore the rates of denitrification measured under oxic conditions can be taken as a conservative minimum.

Third, we used oxygen consumption as a proxy for respiration in these sediments. It has previously been shown that ~50% of oxygen consumption in sediments can be driven by the oxidation of reduced solutes (Cook et al., 2007). In this case, however, we believe that this was unlikely because, we waited >14 hours before oxygen consumption measurements commenced, after which we would expect all the reduced solutes to have either been washed out, or oxidized. We therefore believe that the vast majority of $O_2$ consumption was respiration as opposed to reduced solute oxidation. Finally, break-through curves are often used to quantify column retention time and dispersivity. In this instance, we did not undertake break-through curve measurements, as we have previously shown this column set up to give a very distinct plug flow

(Evrard et al., 2013).  The offset of 10 minutes between the purging of oxygen in the reservoir and the response at the column outlet qualitatively confirms that the theoretical retention time of the columns for these experiments.

## 4.2 Comparison of potential denitrification rates with previous studies

The denitrification rates measured in the present study spanned the range of flow through reactor rates rates ($\sim<1 - 32$ µmol
$L^{-1}$ $h^{-1}$) previously observed in silicate sands (Evrard et al., 2013; Kessler et al., 2012; Marchant et al., 2014; Rao et al., 2007).  The availability, and composition, of organic matter is expected to be a key factor controlling potential denitrification rates (Eyre et al., 2013a; Seitzinger, 1988), and the importance of this in permeable sediments has also been recently underscored by Marchant et al. (2016) who observed a strong relationship between potential denitrification rate and sediment oxygen consumption.   If the results of the previous studies are plotted versus sediment oxygen consumption rate, a
significant relationship is observed with an $r^2$ of 0.92 (Figure 6).  Sites 2 and 3 in the present study seemed to deviate significantly from this relationship, as they were the only data points to lie outside the 99% prediction interval, while site 1 sat close to the line of best fit.  Omitting the study of Kessler et al. (2012), still led to sites 2 and 3 being outside the prediction intervals with site 2 being below, and site 3 above the relationship observed for silicate sands.  This suggests that the slope of the relationship between denitrification and oxygen consumption rate in this study differs from previous studies.
It has recently been shown that much of the metabolism in permeable sediments is dominated by algae, rather than bacteria (Bourke et al., 2017).  Given that bacteria undertake denitrification it is likely that there was relatively more algal respiration occurring at site 2. We speculate that site 2 which was the most sheltered and had the finest sediment was dominated by microphytobenthos, while site 3, which had coarser sediment and higher turbulence in the outer lagoon had a larger advective supply of phyto-detritus from the water column owing to higher flushing rates (Huettel and Rusch, 2000).


## 4.3 Denitrification within carbonate sand grains

The experiments performed here showed denitrification was able to take place at oxygen concentrations oxygen concentrations below 20 µM at site 1 in the FTR and SR experiments and as high as 50 µM in the coarse fraction in the SR experiments (Figures 4 and 5).   It has previously been shown that nanomolar concentrations of oxygen can inhibit
denitrification (Dalsgaard et al., 2014), and one possible explanation is that denitrification was taking places within anoxic niches within the grains.  Theoretically, the critical radius (r) of a particle at which anoxia will occur in the centre can be calculated from equation 3 (Jørgensen, 1977):

$$r = (6D_sC/J)^{1/2} \hspace{4cm} (2)$$

where $D_s$ is the diffusion coefficient (corrected for tortuosity), C is the oxygen concentration at the particle surface and J is
the volumetric oxygen consumption rate. At sediment respiration rates of $270 - 460$ µmol $L^{-1}$ $h^{-1}$ observed in this study, we would expect to see the centre of particles ~1 mm in diameter become anoxic only at oxygen concentrations <5 µM, particles at 2 mm diameter to become anoxic at oxygen concentrations of ~10-20 µM and particles >4mm to be anoxic <50 µM $O_2$. Given that ~20 - 30% of the particles fell in this size range 1-3 mm, we would expect denitrification to commence at $O_2$ concentrations of ~10-20 µM if significant rates of denitrification were taking place within the particles, which is consistent
with our findings for site 1.  The finding that denitrification could take place in the coarse fraction at oxygen concentrations as high as 50 µM is also consistent with this, as this size class encompassed the range ~2-10 mm.

In addition to denitrification taking places within anoxic micro-niches, it is also possible that denitrification under oxic conditions is occurring as has been previously reported (Gao et al., 2012).  We believe this is unlikely for the following
reasons.  Firstly, no denitrification was detected under bulk oxic conditions in the finest sediment at site 2, suggesting

organisms responsible for oxic denitrification were not active in permeable carbonate sediments. Second, oxic denitrification rates were higher in the FTR compared to the SR experiments. In the SR experiments, a maximum oxic denitrification rate ($O_2 > 1$ μM) of 1.4 μmol N $L^{-1}$ $h^{-1}$ (30% of the anoxic rate) was observed at site 1, which compares to a maximum oxic denitrification rate of 4.6 μmol N $L^{-1}$ $h^{-1}$ (45% of the anoxic rate) at site 1 in the FTRs. In the FTRs, there will be a much thicker boundary layer limiting the diffusion of oxygen into the grains than in SRs, and hence a greater anoxic volume and hence denitrification rate than in the stirred reactors. This observation cannot easily be explained by the presence of true 'oxic' denitrification. We note that our maximum rate of denitrification under bulk oxic condition measured in the FTR reactors (4.6 μmol N $L^{-1}$ $h^{-1}$) is at the lower end of oxic denitrification rates reported in silicate sands of ~6 – 17 μmol N $L^{-1}$ $h^{-1}$ (Gao et al., 2010; Marchant et al., 2017), suggesting 'oxic' denitrification, where it occurs, has a greater enhancement effect on total denitrification and denitrification in anoxic microniches.

**4.4 Implications for nitrogen cycling**

Our data suggest that potential anoxic denitrification rates in tropical carbonate sands are low compared to total respiration rates as has previously been observed in silicate sands. We also found evidence to support the hypothesis that denitrification within porous sand grains was taking place. Using a simulation model of a flow field within a ripple, it has previously been shown that in the absence of any intra-granular porosity effect, potential denitrification rates in the range of those measured here scale up to only ~5 μmol $m^{-2}$ $h^{-1}$, however under the same conditions with intra-granular porosity, rates increased by an order of magnitude to 50 μmol $m^{-2}$ $h^{-1}$ owing to intense coupled nitrification denitrification within the sediment grains near the oxic anoxic interface (Kessler et al., 2014). Previous chamber measurements for sites 1 and 2, typically show denitrification rates on the order of 60 μmol $m^{-2}$ $h^{-1}$ and no significant difference was observed between the two sites (Eyre et al., 2013b). It is therefore possible that intra-granular porosity can explain some of the discrepancy between the chamber and modelled results, particularly at site 1. At site 2, however, where we saw no evidence for denitrification under oxic conditions, this is less clear. One possible explanation is that the chambers incorporate a large volume of sediment, which will may include larger grains than the small subsample used in reactor experiments. We also note that for the experiments where we did observe denitrification under bulk oxic conditions, the rates were highly variable, which may suggest a subset of sediment grains (shell versus coral derived), or possibly organisms such as foraminifera (Risgaard-Petersen et al., 2006) may play a disproportionate role in denitrification. Under this scenario it is also possible that the chamber experiments at site 2 enclosed these sediment types, which may have been excluded by chance in the relatively small volume of sediment used in the reactor experiments. Another possible reason for the discrepancies between the chamber and modelled rates are artefacts associated with using chambers in permeable sediments. Model simulations have shown that there may be 'wash out' of nitrogen accumulated within porewaters which can enhance measured nitrogen fluxes (Cook et al., 2006) possibly explaining the higher chamber rates. Although strong relationships between respiration and denitrification in permeable carbonate sands measured using chambers suggests the denitrification rates are reliable (Eyre et al., 2013).

Overall, these results suggest that denitrification may take place within anoxic sites in porous carbonate grains with porewater $O_2$ concentrations of up to 50 μM. The broader ecological significance of this however remains to be elucidated. We suggest further studies be undertaken to: 1. Investigate the effect of different grain types (coral versus shell derived) and sizes and organisms (e.g. foraminifera) on denitrification. 2. Investigate the extent of coupling between nitrification and denitrification within carbonate grains. 3. The use of flume experiments in combination with [15]N tracers to experimentally test the extent of enhanced denitrification under realistic flow fields.

## 5. Author contributions

All authors contributed to the design, undertaking the experiments, data interpretation and manuscript preparation.

## 6. Acknowledgements

This work was supported by the Australian Research Council grants DP150102092 and DP150101281 to BDE and PLMC respectively. We thank Vera Eate for analysis of $^{15}$N-N$_2$ and Michael Bourke for assistance in the laboratory. We thank 5 anonymous reviewers whose comments have helped improve this manuscript.

260

**Table 1 shows sediment grain size, permeability, grain porosity and sediment oxygen consumption rate (with S.D.) at the 3 study sites. $O_2$ consumption and denitrification rates are from flow through reactor experiments**

| Site | Coordinates | Median grain size mm | % 1.18 - 3 mm | % 0.5 – 1.18 mm | % 0.125 – 0.5 mm | % <0.124 mm | Permeability $\times 10^{-12}$ $m^2$ | Bulk Sediment porosity | Grain Porosity Vol/vol | $O_2$ Consumption $\mu$mol $L^{-1}$ $h^{-1}$ | Vmax denitrification rate $\mu$mol $L^{-1}$ $h^{-1}$ |
|------|-------------|------|------|------|------|------|------|------|------|------|------|
| 1 | 23° 26' 37''S 151° 54' 46'' E | 0.9 | 32 | 54 | 13.4 | 0.6 | 27 | 0.55 | 0.31 | 350 (50) | 11.2 (0.3) |
| 2 | 23° 26' 37'' S 151° 55' 09''E | 0.7 | 20 | 44 | 29 | 8 | 24 | 0.48 | N/A | 270 (80)* | 2 (0.5) |
| 3 | 23° 27' 04'' S, 151° 57' 28'' E | 0.9 | 32 | 52 | 16 | 1 | 30 | 0.56 | 0.32 | 466 (9) | 28 (2) |

N/A = not analysed, *n= 2, range shown

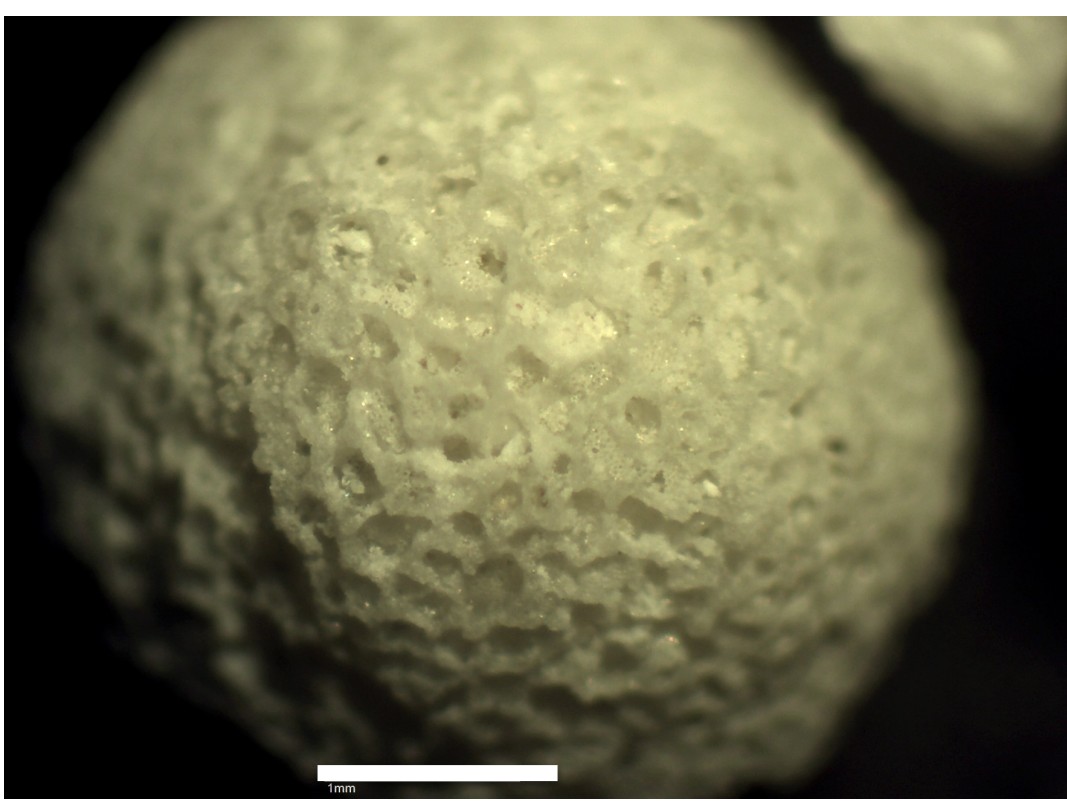

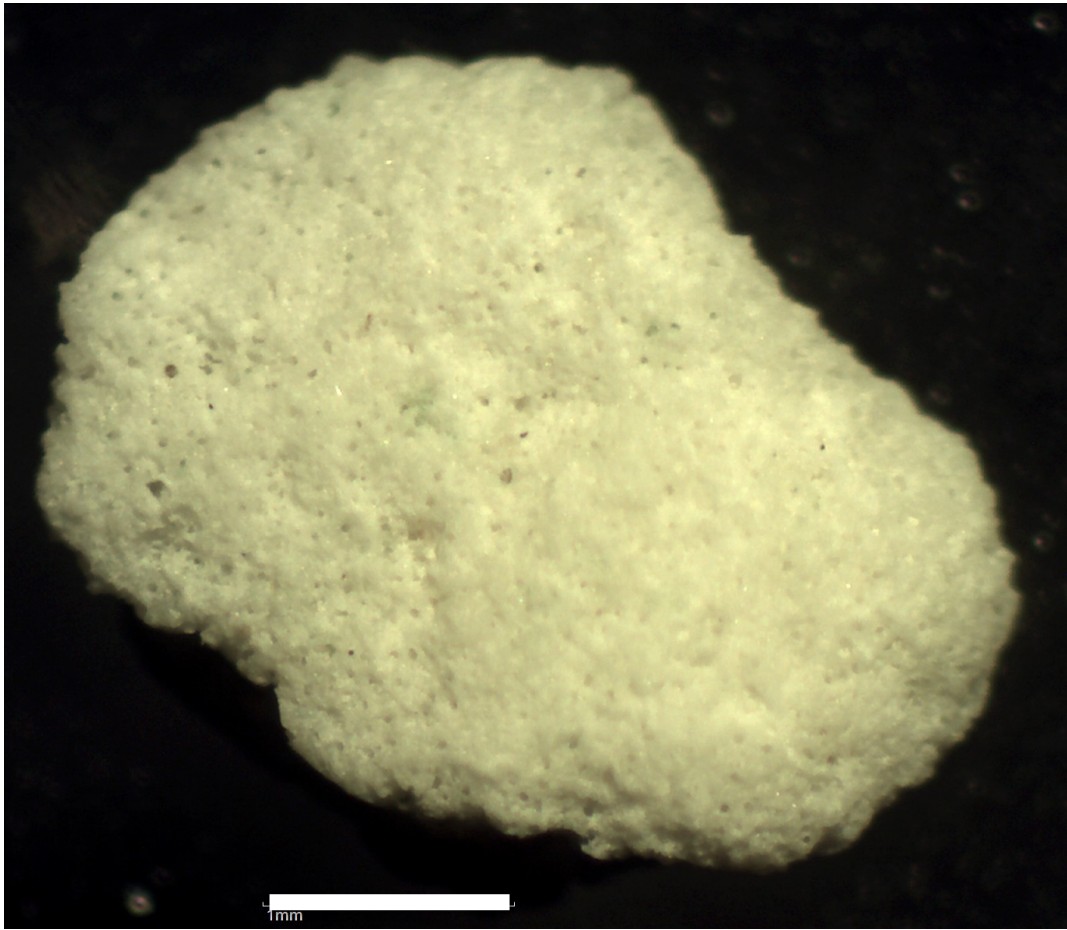

**Figure 1.  Images of carbonate grains from Site 3 at Heron Island.  Scale bar is 1mm.**

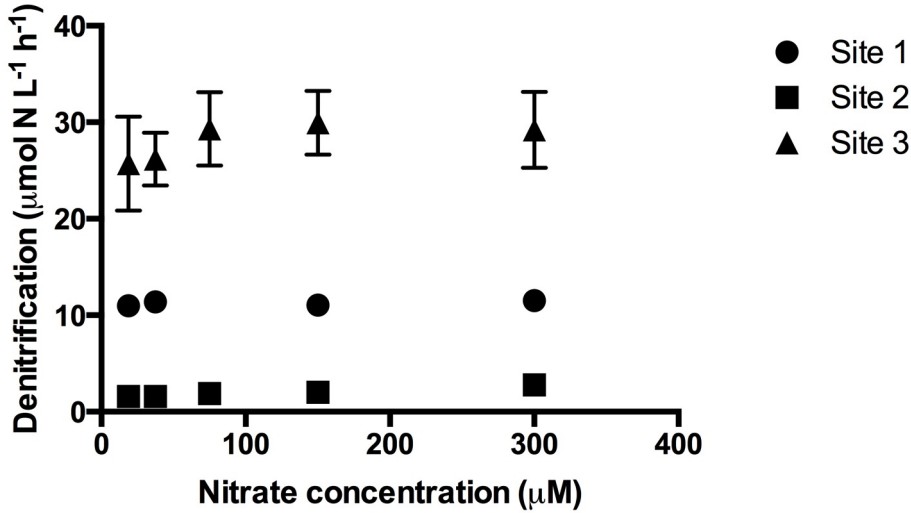

**Figure 2. Denitrification rates as a function of nitrate concentration at the inlet of the columns at the 3 sites studied. Error bars for sites 1 and 2 are smaller than the marker symbols.**


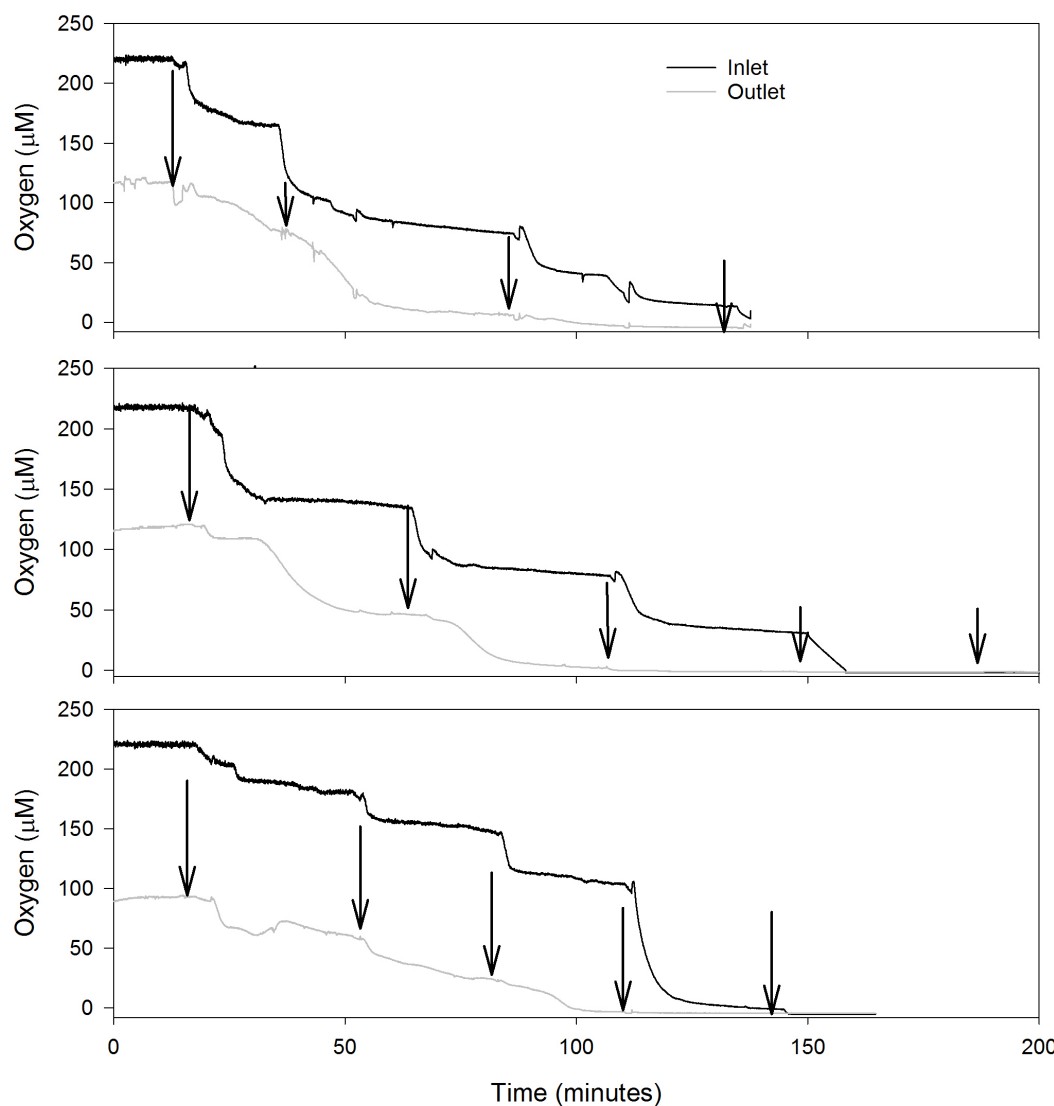

**Figure 3. Examples time series of oxygen concentrations at the FTR inlet and outlet at sites 1 (top panel) 2 (middle panel) and 3 (bottom panel). Arrows mark sampling points for $^{15}$N-N$_2$, note $^{15}$NO$_3^-$ tracer was added > 30minutes before the first sampling point (ie before the O$_2$ trace commences).**


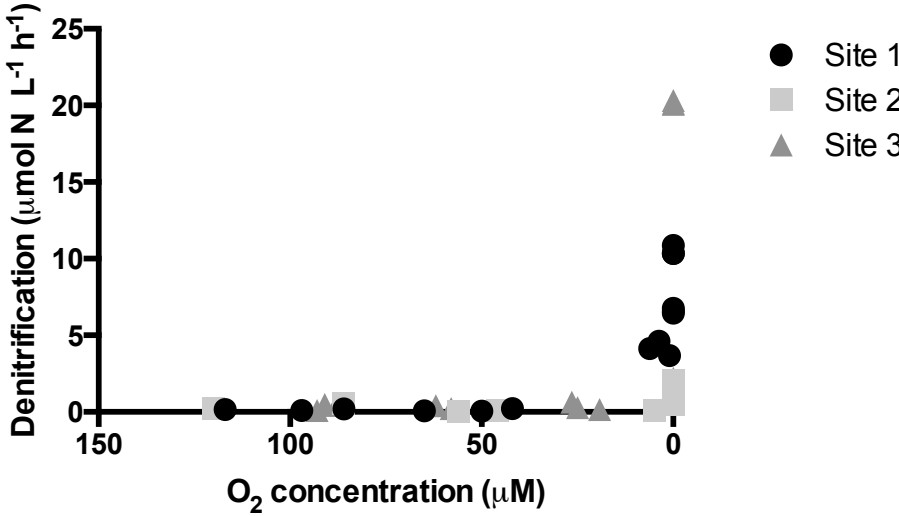

Figure 4. Denitrification rate as a function of oxygen concentration at the outlet of the columns at the 3 study sites.

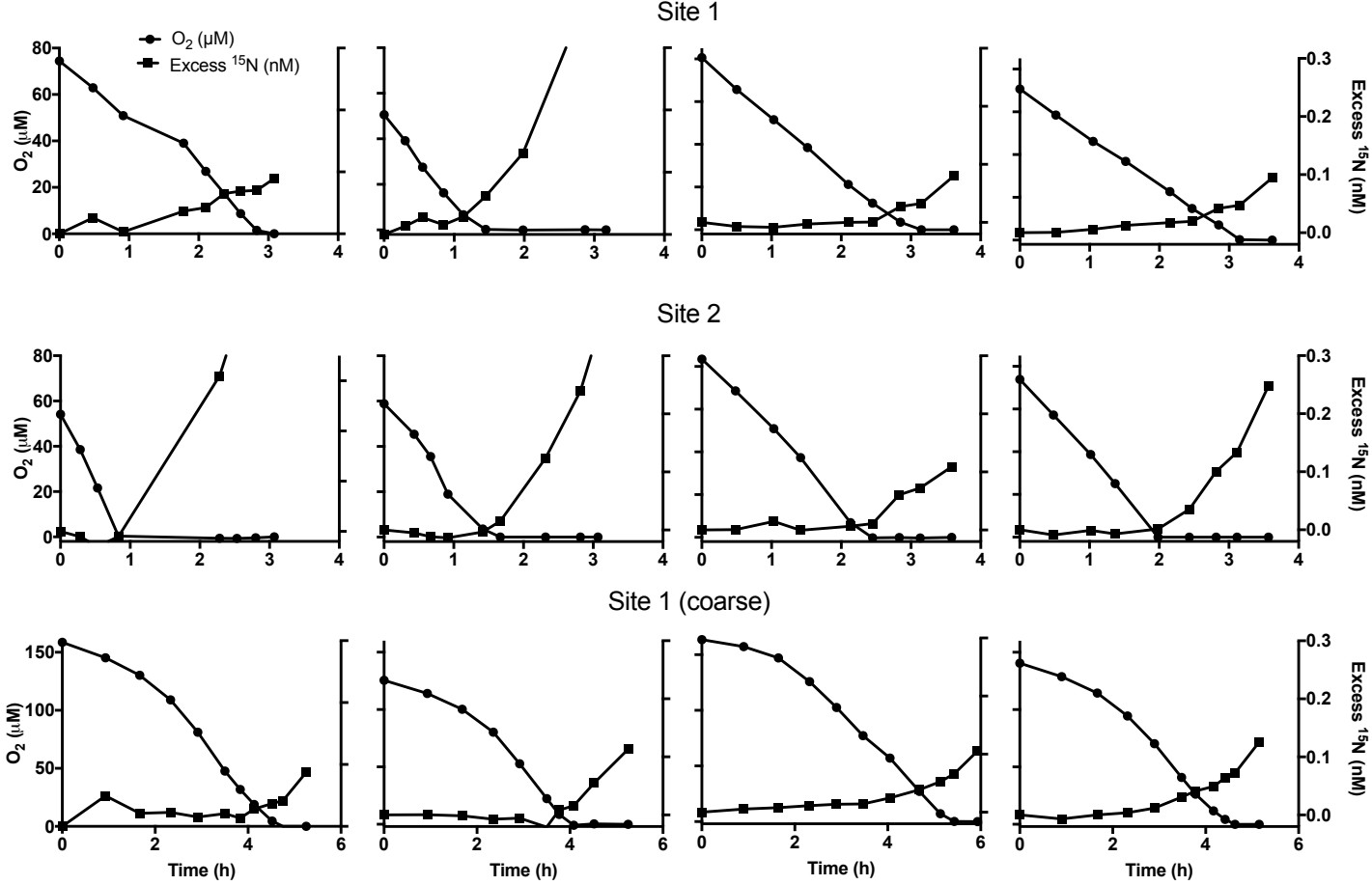

**Figure 5.** **Time series of oxygen and excess $^{15}$N-N$_2$ in stirred reactor experiments on sediments taken from Sites 1 (top panels) and 2 (middle panels) as well as the coarse fraction of sediment (>2 mm) collected from site 1 (bottom panels). Each row shows four replicate experiments for each site.**

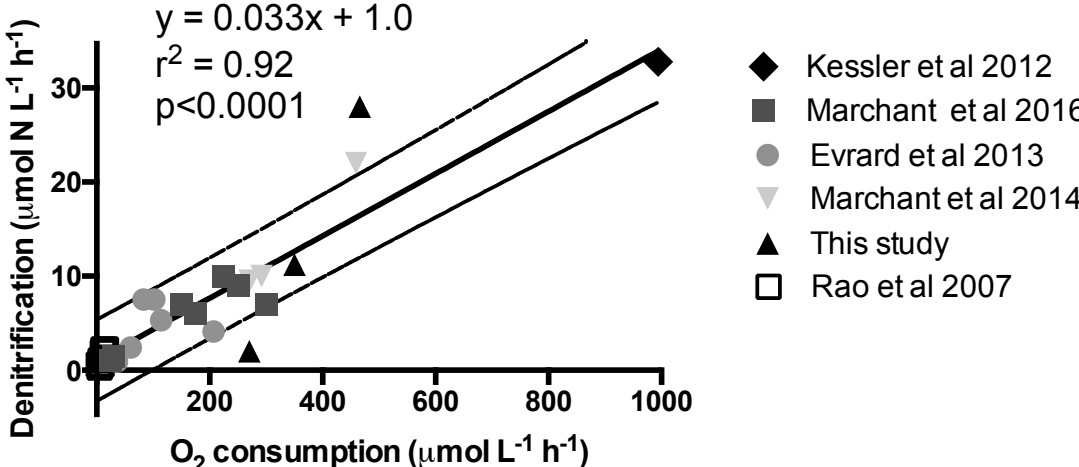

**Figure 6. Denitrification rate as a function of oxygen consumption in this, and previously published studies. The solid line is the line of best fit for previous studies in silicate sands, the dashed lined are the 99% prediction interval. Data from this study are from the flow through reactor experiments.**

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
