# Peer review of "Does denitrification occur within porous carbonate sand grains?"

_Biogeosciences, 2016_

## Referee Comment (RC1) · Anonymous Referee #1 · 2 Feb 2017

The manuscript by Cook et al discusses, based on flow-through reactor experiments, the possibilities for denitrification in porous sand grains, which may possibly act as oxygen depleted microniches thus providing an environment for oxygen-sensitive denitrification. The authors, however, conclude that there is no evidence for the existence of those microniches based on the oxygen sensitivity and the absence of diffusion limitation for nitrate. Given the current microniche-focus in N cycle research, Cook et al. add an interesting piece of work which should be available to the readers of Biogeosciences. To me the manuscript is overall very interesting, nicely structured and focused to the point. I have several comments listed in the following which I believe have to be addressed to make it clearer to the reader:

General comments:

[Figure]

1. Denitrification is defined, in the beginning, however, other N cycle processes are not mentioned in this context (anammox, DNRA), although the methods part basically describes that at least anammox-derived N2 would have been measurable. Both of them would in principle be able to occur under similar conditions.

2. To me, the fact that denitrification occurs already at $10\mu$M O2 is rather an indication that there is indeed a bit of an effect on the process. $10\mu$M O2 is pretty high for measurable rates of N2 production. Dalsgaard et al (2014) actually showed that minimal changes in O2 can largely impact on denitrification rates

3. I would like to see more of a discussion of what this means globally, are there many sediments like this which were suspected to be sites of intense N loss? This basically requires to make a stronger statement on your results. In this context, those grains may not act as microniches for denitrification, however, there may be tipping points e.g. if the organic carbon source increases by eutrophication where indeed this changes. What I think of is that maybe they are just not microniches, yet.

T Dalsgaard et al., Oxygen at Nanomolar Levels Reversibly Suppresses Process Rates and Gene Expression in Anammox and Denitrification in the Oxygen Minimum Zone off Northern Chile mBio 5 (6), e01966-14, 2014.

Specific comments:

Throughout the text: Please check, whether abbreviations are spelled out when mentioned the first time, please unify O2/ oxygen, please check units (sometimes it says uM instead of $\mu$M)

p. 2, l18: Please clarify, which process rates.

l. 20: I disagree on that statement, $10\mu$M are pretty high for full denitrification.

L23, Please remove 'rates' after denitrification.

l. 23: I am missing a sentence on the meaning of the result, here.

l. 31: This effect could be positive or negative, could you elaborate a bit more?

l. 36 Insert 'under' after 'place'.

p. 3, l 57: Where does this number ($50\mu$M) come from?

l. 70 ff: I would like to see a map with the sampling locations. Also, for all companies, a location should be added.

l.95: This also changes the $CO_2$ content and with that the pH, what could be the impact?

l. 117: This sentence is odd, please rephrase

l. 163: This doesn't necessarily have to happen, it may be that denitrification occurs in a range where it wouldn't occur without porous grains.

l. 184: The fact that the anoxic zone is reduced doesn't necessarily translate into lower denitrification in a case where substrate supply is higher. It may actually well be that at the oxycline a zone of intense denitrification forms.

l. 190: remove 'rates'

l.201: Does this make sense in your sediments in terms of light penetration depths?

l. 216: Could also be over-estimating- could you add a reference so that it gets clear what you are talking about, here?

I wish the conclusion could end with a stronger statement on the meaning of your results.

---

## Referee Comment (RC2) · Anonymous Referee #2 · 6 Feb 2017

The authors test the hypothesis whether micro-sites in porous carbonate sands can become anoxic, thus providing important niches for denitrification under bulk oxic conditions. They use flow through reactors (FTRs) packed with carbonate sands from 3 station and measured the denitrification rates under various oxygen and nitrate concentrations, postulating that any diffusion limitation of O2 or NO3- in the micro-niche should be observable in the bulk denitrification rates. The authors measured very different O2 and NO3- consumption rates at the 3 stations, but they observed no change of denitrification rates at each site for decreasing NO3- concentrations down to 18 $\mu$M (the lowest inflow concentration tested). Under bulk oxic conditions, denitrification rates were only measured when the outflow O2 was below 10 $\mu$M. The authors suggested that anoxic micro-niches do not exist and that denitrification is not affected.

The authors address an important problem in sediment biogeochemistry which is still

not resolved: whether denitrification is active in permeable sands under bulk oxic conditions. The manuscript is well organized and clearly written. I have, however, some major concerns about the proposed interpretation of the results.

The authors use flow through reactors (FTRs) to investigate the effect of diffusion limitation on oxic respiration and the formation of anoxic micro-niches, and subsequently on denitrification rates. In general, diffusive transport depends on concentration gradients and such 'limitation experiments' should therefore have full control of the ambient O2 and NO3- concentrations. I doubt that FTRs are the right choice for such experiments, because they produce a considerable concentration gradient between inflow and outflow, which is actually necessary to determine the reaction rate. The differences in O2 concentration at inflow and outflow are well documented in Fig. 3 where they are of the order of a 50-100$\mu$M.

This has some significant implications: when the authors state that (abstract) "denitrification was only observed to commence at substantial rates below 10 $\mu$M O2" they refer to outflow concentrations. This means that O2 concentrations at the inflow must have been between 55$\mu$M and 90$\mu$M (back calculated from O2 rates in table 1 and 10min retention time), so that spatially averaged concentrations in the FTRs are 30-50$\mu$M.

Now, from this perspective, the results actually do support the hypothesis of denitrification in anoxic micro-niches. This is also in line with the authors who conclude from their calculation of diffusion limitation: (equation 3, Line 171) "...we would expect denitrification to have commenced at O2 concentrations below 30-50 $\mu$M... (in case of anoxic micro niches)".

Another argument put forward was the non-limitation of denitrification rates at decreasing NO3- concentrations from 300$\mu$M down to 18$\mu$M (anoxic conditions). Of course, there is also a NO3- gradient in the FTR as described above for O2 which makes it complicated to study such concentration limitation. Further, when applying equation 3, the expected NO3- concentration gradient in a 3mm grain is only 0.3-4$\mu$M – a change

which is probably too little to be reflected in decreased rates.

In summary, it is possible to interpret the results just as well in favor of micro-niche denitrification. In general, I feel that FTRs are not well suited to study concentration dependent rates, because they provide a large variety of different concentrations between in- and outflow. The situation is even worse when considering the dispersion effects of a non-ideal plug flow, which was not discussed at all.

Because concentration differences between in- and outflow are necessary for the rate calculation in FTR studies they cannot be minimized without increasing the error of the rate calculations. A possible way out of this dilemma would be the use of stirred slurry incubations to study concentration dependent rates (as described for example in Gao et al. 2009, ISME doi:10.1038/ismej.2009.127).

Minor comments:

Line 36: For denitrification to take place in (!) anoxic conditions

Line 62: what units have 'a' and 'J' ? J is probably not a flux here. . ...

Line 76: please specify the dimensions of the FTRs.

Line 80: "For denitrification to take place anoxic conditions. . ....." this sentence does not fit here. . .

Line 87: please specify in this section if the measured rates are per volume porewater or per volume wet sediment

Line 113: the permeability is very low for such coarse grain sizes (median 0.9 and 0.7mm). I would expect something in the range of X*10-10 m$^2$. Do you have an explanation?

Line 144: ". . .waited >14 hours before oxygen consumption measurements commenced. . ." Please specify if the cores were flushed and which volume flow you used.

---

## Author Comment (AC1) · 3 Mar 2017

We thank the reviewer for their thoughtful comments

Addressing the 3 general comments

1. Regarding anammox, there was very little evidence for this, given that there was virtually no 29N2 production. This is consistent with previous measurements in sands. This will be addressed in a revised manuscript. 2. The reviewer makes a good point and this is also picked up by the second reviewer. We will broaden the discussion to take this into account and also conduct some additional experiments (see response to reviewer 2). 3. The discussion will be broadened to incorporate the global implications of these findings into the manuscript.

[Figure]

The other minor specific comments will be addressed in the revised manuscript

---

## Author Comment (AC2) · 3 Mar 2017

We thank the reviewer for their thoughtful comments

This reviewer makes a good point that that stirred slurry experiments would make our findings more convincing. We agree with this and will undertake additional slurry experiments with samples collected from two of the three study sites in March. The results of this will be incorporated into the revised manuscript to make the conclusions more robust.

Other minor comments will be addressed in the revised manuscript.

---

## Author Response (AR1)

We thank both reviewers for their constructive criticism of the manuscript. Both reviewers raise the point that the data may actually support the hypothesis that denitrification occurs within anoxic grains. Furthermore, reviewer 2 suggests additional stirred reactor experiments to test this hypothesis. We have now undertaken these and the results do indeed suggest denitrification occurs under bulk oxic conditions. On this basis, we have completely re-written the manuscript. Below is our detailed response to the reviewer comments (in bold).

Reviewer 1

The manuscript by Cook et al discusses, based on flow-through reactor experiments, the possibilities for denitrification in porous sand grains, which may possibly act as oxygen depleted microniches thus providing an environment for oxygen-sensitive den- itrification. The authors, however, conclude that there is no evidence for the existence of those microniches based on the oxygen sensitivity and the absence of diffusion lim- itation for nitrate. Given the current microniche-focus in N cycle research, Cook et al. add an interesting piece of work which should be available to the readers of Bio-geosciences. To me the manuscript is overall very interesting, nicely structured and focused to the point. I have several comments listed in the following which I believe have to be addressed to make it clearer to the reader:

General comments:

1. Denitrification is defined, in the beginning, however, other N cycle processes are not mentioned in this context (anammox, DNRA), although the methods part basically describes that at least anammox-derived N2 would have been measurable. Both of them would in principle be able to occur under similar conditions.

**We did not quantify DNRA, so an exact measurement of anammox is not possible with our data, but we can make an estimate if we assume this process to be negligible. Using the 29/30 ratio from the 300 $\mu$M 15NO3- addition, we can conservatively assume that all the 29N2 production is due to anammox. Using equation 23 from Risgaard-Petersen et al, we calculate anammox comprised a maximum of 16% of N2 production.**

**We have now briefly mentioned this in the methods section, and we now state that anammox comprised <16% of N$_2$ production in the results**

2. To me, the fact that denitrification occurs already at 10$\mu$M O2 is rather an indication that there is indeed a bit of an effect on the process. 10$\mu$M O2 is pretty high for mea- surable rates of N2 production. Dalsgaard et al (2014) actually showed that minimal changes in O2 can largely impact on denitrification rates

**In response to both reviewers comments on this, we have undertaken addition stirred reactor experiments and now completely re-interpreted our results along the lines suggested here. Of particular relevance to this comment, we now start section 4.3 with the following sentences**

**'The experiments performed here showed denitrification was able to take place at oxygen concentrations oxygen concentrations below 20 $\mu$M at site 1 in the FTR and SR experiments and as high as 50 $\mu$M in the coarse fraction in the SR experiments (Figs 4 and 5). It has previously been shown that nanomolar concentrations of oxygen can inhibit**

**denitrification (Dalsgaard et al., 2014), suggesting that denitrification was taking places within anoxic niches within the grains.'**

3. I would like to see more of a discussion of what this means globally, are there many sediments like this which were suspected to be sites of intense N loss? This basically requires to make a stronger statement on your results. In this context, those grains may not act as microniches for denitrification, however, there may be tipping points e.g. if the organic carbon source increases by eutrophication where indeed this changes. What I think of is that maybe they are just not microniches, yet.

**As mentioned above, we have now completely re-interpreted our results. We believe there is evidence for microniches and the ecological implication for this are now discussed in the revised section 4.4. We have avoided any global sttements, as we believe further work is needed before this bigger picture implications are known.**

T Dalsgaard et al., Oxygen at Nanomolar Levels Reversibly Suppresses Process Rates and Gene Expression in Anammox and Denitrification in the Oxygen Minimum Zone off Northern Chile mBio 5 (6), e01966-14, 2014.

Specific comments:

Throughout the text: Please check, whether abbreviations are spelled out when men- tioned the first time, please unify O2/ oxygen, please check units (sometimes it says uM instead of μM)

**Checked and changed as suggested**

p. 2, l18: Please clarify, which process rates.
l. 20: I disagree on that statement, 10μM are pretty high for full denitrification. L23, Please remove 'rates' after denitrification.
l. 23: I am missing a sentence on the meaning of the result, here.

**Abstract now re-written taking these points into account**

l. 31: This effect could be positive or negative, could you elaborate a bit more?

**This gets complicated and is still under debate, we have just said it can both enhance and reduce denitrification.**

l. 36 Insert 'under' after 'place'.

**Changed**

p. 3, l 57: Where does this number (50μM) come from?

**We have now removed specific value and simply stated under low oxygen conditions**

l. 70 ff: I would like to see a map with the sampling locations. Also, for all companies, a location should be added.

**We believe a map is not necessary, there are many previous studies of Heron island with maps and with the advent of Google maps, and the coordinates, an interested reader can instantly look up the sites.**

l.95: This also changes the $CO_2$ content and with that the pH, what could be the impact?

**This would increase the pH, which could change nitrification, however there are no studies that have shown an effect of pH on denitrification to our knowledge. We have been undertaking similar studies for many years now, and we typically see constant denitrification rates over hours of purging (Evrard et al., 2013)over which time, there would be the greatest pH change.**

l. 117: This sentence is odd, please rephrase

**Now rephrased to**

**Rates of denitrification were constant above $NO_3^-$ concentrations of 18 $\mu$M at all three study sites, and were highest at site 3 which had the highest sediment oxygen consumption rates and lowest at site 2 which had the lowest oxygen consumption rates (Figure 2)**

l. 163: This doesn't necessarily have to happen, it may be that denitrification occurs in a range where it wouldn't occur without porous grains.

**This argument has been removed**

l. 184: The fact that the anoxic zone is reduced doesn't necessarily translate into lower denitrification in a case where substrate supply is higher. It may actually well be that at the oxycline a zone of intense denitrification forms.

**This part of the discussion has been removed**

l. 190: remove 'rates'

**We believe this wording is correct, left as is**

l.201: Does this make sense in your sediments in terms of light penetration depths?

**This argument is based not on light penetration, but on the continual mixing of permeable sediment leading to the burial of algae**

l. 216: Could also be over-estimating- could you add a reference so that it gets clear what you are talking about, here?

**This part of the discussion has been substantially revised so as to make this comment obsolete**

I wish the conclusion could end with a stronger statement on the meaning of your results.

**We have now re-written the final section. We have however avoided strong statements as further work is required to investigate the significance of this.**

Reviewer 2

The authors test the hypothesis whether micro-sites in porous carbonate sands can become anoxic, thus providing important niches for denitrification under bulk oxic con- ditions. They use flow through reactors (FTRs) packed with carbonate sands from 3 station and measured the denitrification

rates under various oxygen and nitrate con- centrations, postulating that any diffusion limitation of O2 or NO3- in the micro-niche should be observable in the bulk denitrification rates. The authors measured very dif- ferent O2 and NO3- consumption rates at the 3 stations, but they observed no change of denitrification rates at each site for decreasing NO3- concentrations down to 18 μM (the lowest inflow concentration tested). Under bulk oxic conditions, denitrification rates were only measured when the outflow O2 was below 10 μM. The authors suggested that anoxic micro-niches do not exist and that denitrification is not affected.

The authors address an important problem in sediment biogeochemistry which is still not resolved: whether denitrification is active in permeable sands under bulk oxic con- ditions. The manuscript is well organized and clearly written. I have, however, some major concerns about the proposed interpretation of the results.

The authors use flow through reactors (FTRs) to investigate the effect of diffusion lim- itation on oxic respiration and the formation of anoxic micro-niches, and subsequently on denitrification rates. In general, diffusive transport depends on concentration gradi- ents and such 'limitation experiments' should therefore have full control of the ambient O2 and NO3- concentrations. I doubt that FTRs are the right choice for such exper- iments, because they produce a considerable concentration gradient between inflow and outflow, which is actually necessary to determine the reaction rate. The differ- ences in O2 concentration at inflow and outflow are well documented in Fig. 3 where they are of the order of a 50-100μM.

This has some significant implications: when the authors state that (abstract) "denitrifi- cation was only observed to commence at substantial rates below 10 μM O2" they refer to outflow concentrations. This means that O2 concentrations at the inflow must have been between 55μM and 90μM (back calculated from O2 rates in table 1 and 10min retention time), so that spatially averaged concentrations in the FTRs are 30-50μM.

Now, from this perspective, the results actually do support the hypothesis of denitrifica- tion in anoxic micro-niches. This is also in line with the authors who conclude from their calculation of diffusion limitation: (equation 3, Line 171) ". . .we would expect denitrifi- cation to have commenced at O2 concentrations below 30-50 μM. . . (in case of anoxic micro niches)".

Another argument put forward was the non-limitation of denitrification rates at decreas- ing NO3- concentrations from 300μM down to 18μM (anoxic conditions). Of course, there is also a NO3- gradient in the FTR as described above for O2 which makes it complicated to study such concentration limitation. Further, when applying equation 3, the expected NO3- concentration gradient in a 3mm grain is only 0.3-4μM – a change which is probably too little to be reflected in decreased rates.

In summary, it is possible to interpret the results just as well in favor of micro-niche denitrification. In general, I feel that FTRs are not well suited to study concentration dependent rates, because they provide a large variety of different concentrations be- tween in- and outflow. The situation is even worse when considering the dispersion effects of a non-ideal plug flow, which was not discussed at all.

Because concentration differences between in- and outflow are necessary for the rate calculation in FTR studies they cannot be minimized without increasing the error of the rate calculations. A possible way out of this dilemma would be the use of stirred slurry incubations to study concentration dependent rates (as described for example in Gao et al. 2009, ISME doi:10.1038/ismej.2009.127).

**We thank the reviewer for their thoughtful comments and suggestions.  We have now undertaken the stirred reactor experiments as suggested.  The results do indeed support the**

**presence of denitrification under oxic conditions and we have now re-written the manuscript to reflect this.**

Minor comments:

Line 36: For denitrification to take place in (!) anoxic conditions

**changed**

Line 62: what units have 'a' and 'J' ? J is probably not a flux here. . ...

**This part has now been removed**

Line 76: please specify the dimensions of the FTRs.

**These details now added. 4.6 cm diameter, 4 cm length**

Line 80: "For denitrification to take place anoxic conditions. . .. . ." this sentence does not fit here. . .

**Possible an error in refence to line number?  I can't see this here.**

Line 87: please specify in this section if the measured rates are per volume porewater or per volume wet sediment

**Per volume wet sediment now specified**

**References cited**

Dalsgaard, T., Stewart, F. J., Thamdrup, B., De Brabandere, L., Revsbech, N. P., Ulloa, O., Canfield, D. E., and DeLong, E. F.: Oxygen at Nanomolar Levels Reversibly Suppresses Process Rates and Gene Expression in Anammox and Denitrification in the Oxygen Minimum Zone off Northern Chile, mBio, 5, e01966-01914, 2014.
Evrard, V., Glud, R. N., and Cook, P. L. M.: The kinetics of denitrification in permeable sediments, Biogeochemistry, 113, 563-572, 2013.

---

## Referee Report (RR1)

**Review of the manuscript 'Does denitrification occur within porous carbonate sand grains?' (bg-2016-530) by Perran Cook et al.**

General comments:

The manuscript 'Does denitrification occur within porous carbonate sand grains?' by Cook et al. addresses the question whether porous carbonate sands provide anaerobic micro niches. The authors use a series of controlled incubation experiments with stirred and flow-through reactors in combination with isotopically labeled nitrate. This design enables to quantify and understand the kinetics of N-turnover in sediments. The appealing detail of the presented study is the use of porous carbonate sand, which potentially provides the disputed micro niches with a high probability. The comparison of N-turnover in sediment composed of either porous or non-porous grains might make an important point in the ongoing debate on micro niches.

The manuscript is well written and structured, the language is better than mine.

However, my main concern is that the authors missed good opportunities to add new insights to the ongoing discussion on the significance of micro niches.

1.) From my perspective, the question is not whether or not such anaerobic micro niches actually occur in natural sediment. There are always sheltered places within poorly connected pores and angles, and grooves and cracks occur even on quartz grains (a tracer break-through curve would reveal the connectedness of the pore network). In comparison with more exposed grain surfaces, these niches have a reduced exchange with the bulk pore water. It is thus inevitable that some spots within a complex pore network become earlier anoxic than the (averaged) oxygen concentration of the bulk pore. Vigorous stirring as performed in the experiments with the stirred reactor should eliminate or at least reduce such inter-granular micro niches caused by grain roughness or poorly connected intergranular pores. So one would expect that denitrification is more sensitive to the bulk oxygen concentration in the stirred reactor than in the packed flow-through reactor. But the presented data indicates the opposite effect: Denitrification in stirred reactor commenced at higher oxygen concentration (section 3), and the authors did not discuss this. It would be helpful here to see and to compare the volumetric denitrification rates (mol N per time and sediment volume) of both reactor types. Ultimately, the comparison of the results from the stirred reactor and porous grains with the results from Gao et al. (2010, doi:10.1038/ismej.2009.127) with non-porous sand could elucidate the effect of the internal porosity of the carbonate sand. Unfortunately, the authors missed this opportunity.

2.) The authors explain the significance of the anaerobic micro niches with the potentially tight coupling between nitrification and denitrification. In this context would it be interesting to see the in-situ concentrations of nitrate and ammonium (the substrate for nitrification). Additionally, the isotope pairing method employed by the authors enables to differentiate between the ambient nitrate (14N) and the labeled nitrate (15N). If the micro niches contribute to the overall N-cycling then I would expect that the contribution of ambient N (nitrate + ammonium) to total denitrification differs between the stirred and flow-through reactors. But unfortunately, the authors did not provide these results. A little more effort (e.g. labeling the ammonium in incubations) or measuring the

isotope ratio of the ambient nitrate (Deek et al. 2013, doi:10.3354/meps10514 ) would directly enable to quantify the coupled nitrification-denitrification.

3.) As the authors underlined in section 4.2 is the 'availability, and composition of organic matter […] a key factor controlling potential denitrification rates'. However, the authors did not present any measurement on quantity or quality of the organic matter such as total organic carbon (TOC), amino-acid based reactivity index (RI) or degradation index (DI). Even a simple measurement of chlorophyll content would provide a proxy for the sediment reactivity, and would also enable to discuss the observed differences in the denitrification rates without having to speculate about phyto-detritus (p. 6, l. 190). Such organic matter proxies as part of a thorough description of the sampling site would also help other researchers to re-use the denitrification rates presented by the authors.

In summary, the presented results could make an important contribution to the understanding of N-cycling in sediment with complex structure and should be published. But I have the impression that the presentation of results and the discussion are incomplete, and that the presented manuscript does not tap its full potential. In its current form it is impossible to conclude whether or not the internal porosity of carbonate grains contributes to denitrification.

I suggest that the authors present and discuss the complete volumetric denitrification rates for both reactor types, preferably as a table. This summary should also give the detailed results of the isotope pairing incubation (D14 and D15 separately, estimates for anammox and nitrification). A more complete characterization of the sampled sites with an organic matter proxy (TOC, DI, RI, chlorophyll) and ambient concentrations of nutrients would increase the re-usability of the results for other topics.

Minor comments:

- Section 2.1, p. 2, l. 83: Reaction rates are were calculated … -> delete 'are'
- Section 2.2, p. 4, l. 105: course -> coarse?
- Table 1: Are the denitrification rates from SR or FTR?
- Figure 2: Are the error bars of site 1 and 2 missing or just too small to show?
- Figure 6: From which type of experiment are the presented results- SR or FTR?
- Figure 6: Error bars (where applicable) are missing

---

## Author Response (AR2)

Response to each reviewer are given in *italic*.  Changes are detailed in **bold**

*We thank both reviewers for their thoughtful comments, we now address these below*

**Response to reviewer 1**

General comments:
The manuscript 'Does denitrification occur within porous carbonate sand grains?' by Cook et al. addresses the question whether porous carbonate sands provide anaerobic micro niches. The authors use a series of controlled incubation experiments with stirred and flow-through reactors in combination with isotopically labeled nitrate. This design enables to quantify and understand the kinetics of N-turnover in sediments. The appealing detail of the presented study is the use of porous carbonate sand, which potentially provides the disputed micro niches with a high probability. The comparison of N-turnover in sediment composed of either porous or non-porous grains might make an important point in the ongoing debate on micro niches.
The manuscript is well written and structured, the language is better than mine.
However, my main concern is that the authors missed good opportunities to add new insights to the ongoing discussion on the significance of micro niches.

1.) From my perspective, the question is not whether or not such anaerobic micro niches actually occur in natural sediment. There are always sheltered places within poorly connected pores and angles, and grooves and cracks occur even on quartz grains (a tracer break-through curve would reveal the connectedness of the pore network). In comparison with more exposed grain surfaces, these niches have a reduced exchange with the bulk pore water. It is thus inevitable that some spots within a complex pore network become earlier anoxic than the (averaged) oxygen concentration of the bulk pore. Vigorous stirring as performed in the experiments with the stirred reactor should eliminate or at least reduce such inter-granular micro niches caused by grain roughness or poorly connected intergranular pores. So one would expect that denitrification is more sensitive to the bulk oxygen concentration in the stirred reactor than in the packed flow-through reactor. But the presented data indicates the opposite effect: Denitrification in stirred reactor commenced at higher oxygen concentration (section 3), and the authors did not discuss this. It would be helpful here to see and to compare the volumetric denitrification rates (mol N per time and sediment volume) of both reactor types.

*We believe that apparent commencement of denitrification at higher oxygen concentrations in the stirred reactors arises as a consequence of two factors. Firstly, it is hard to compare the two approaches, because the rate of denitrification in an FTR encompasses a wide range of oxygen concentrations.  When denitrification commences under oxic conditions, it is at very low rates.  Within the stirred reactors, the small increase in 15N-N2 can be detected, however in the flow through reactors, only a very small volume of the reactor will be in the low oxygen region where denitrification occurs and hence the rates when expressed per volume of reactor will be extremely low.  Second, there are only a small number of data points where the oxygen concentration at the outlet is >0 but within the concentration range where denitrification was observed to commence in the FTRs, making an assignment of the exact oxygen threshold impossible.*

*At site 1, we do have one point where the oxygen concentration at the outlets is 6 µM and the corresponding denitrification rate is 4 µmol N L$^{-1}$ h$^{-1}$ (40% of the anoxic rate), which is higher than 1.3 µmol N L$^{-1}$ h$^{-1}$ (30% of the anoxic rate) measured the stirred reactor experiments.*

*We have now added the following discussion*

**In addition to denitrification taking places within anoxic micro-niches, it is also possible that denitrification under oxic conditions is occurring as has been previously reported (Gao et al., 2012). We believe this is unlikely for the following reasons. Firstly, no denitrification was detected under bulk oxic conditions in the finest sediment at site 2, suggesting organisms responsible for oxic denitrification were not active in permeable carbonate sediments. Second, oxic denitrification rates were higher in the FTR compared to the SR experiments. In the SR experiments, a maximum oxic denitrification rate (O$_2$ > 1 µM) of 1.4 µmol N L$^{-1}$ h$^{-1}$ (30% of the anoxic rate) was observed at site 1, which compares to a maximum oxic denitrification rate of 4.6 µmol N L$^{-1}$ h$^{-1}$ (45% of the anoxic rate) at site 1 in the FTRs. In the FTRs, there will be a much thicker boundary layer limiting the diffusion of oxygen into the grains than in SRs, and hence a greater anoxic volume and hence denitrification rate than in the stirred reactors. This observation cannot easily be explained by the presence of true 'oxic' denitrification. We note that our maximum rate of denitrification under bulk oxic condition measured in the FTR reactors (4.6 µmol N L$^{-1}$ h$^{-1}$) is at the lower end of oxic denitrification rates reported in silicate sands of ~6 – 17 µmol N L$^{-1}$ h$^{-1}$ (Gao et al., 2010; Marchant et al., 2017), suggesting 'oxic' denitrification, where it occurs, has a greater enhancement effect on total denitrification and denitrification in anoxic microniches.**

Ultimately, the comparison of the results from the stirred reactor and porous grains with the results from Gao et al. (2010, doi:10.1038/ismej.2009.127) with non-porous sand could elucidate the effect of the internal porosity of the carbonate sand. Unfortunately, the authors missed this opportunity.

*The last 6 lines of the new discussion in the previous response address this point.*

2.) The authors explain the significance of the anaerobic micro niches with the potentially tight coupling between nitrification and denitrification. In this context would it be interesting to see the in-situ concentrations of nitrate and ammonium (the substrate for nitrification). Additionally, the isotope pairing method employed by the authors enables to differentiate between the ambient nitrate (14N) and the labeled nitrate (15N). If the micro niches contribute to the overall N-cycling then I would expect that the contribution of ambient N (nitrate + ammonium) to total denitrification differs between the stirred and flow-through reactors. But unfortunately, the authors did not provide these results. A little more effort (e.g. labeling the ammonium in incubations) or measuring the isotope ratio of the ambient nitrate (Deek et al. 2013, doi:10.3354/meps10514 ) would directly enable to quantify the coupled nitrification-denitrification.

*We have carefully reviewed our data and indeed the 14N/15N ratio in the N2 gas is much higher than the 14N/15N ratio in the starting nitrate pool (based on the ratio of 15NO3*

*added compared to the native 14NO3 pool) indicating a large amount of nitrification. We did not measure the 14N/15N ratio of the nitrate pool over time in the bulk porewater pool, so we are not able to calculate the additional nitrification coupled to denitrification taking place within the grains. Indeed, we had suggested this as an important avenue for future research. No change has been made.*

2.) As the authors underlined in section 4.2 is the 'availability, and composition of organic matter [...] a key factor controlling potential denitrification rates'. However, the authors did not present any measurement on quantity or quality of the organic matter such as total organic carbon (TOC), amino-acid based reactivity index (RI) or degradation index (DI). Even a simple measurement of chlorophyll content would provide a proxy for the sediment reactivity, and would also enable to discuss the observed differences in the denitrification rates without having to speculate about phyto-detritus (p. 6, l. 190). Such organic matter proxies as part of a thorough description of the sampling site would also help other researchers to re-use the denitrification rates presented by the authors.

*This discussion pertains to phytodetritus versus microphytobenthos. Both of these sources of organic matter have essentially the same composition, so these measurements would not add anything to the interpretation. The key difference between these two factors is that one is living (microphytobenthos) and hence creates competition for nitrogen and one is dead (phytodetritus) and is hence a source of nitrogen). No change has been made*

In summary, the presented results could make an important contribution to the understanding of N-cycling in sediment with complex structure and should be published. But I have the impression that the presentation of results and the discussion are incomplete, and that the presented manuscript does not tap its full potential. In its current form it is impossible to conclude whether or not the internal porosity of carbonate grains contributes to denitrification.

*We disagree, we have shown that oxic denitrification was observed in the sediments with coarser grain size but not in the finer sediments from site 2. As discussed in the manuscript, we believe this provides strong evidence that denitrification occurs within the grains.*

I suggest that the authors present and discuss the complete volumetric denitrification rates for both reactor types, preferably as a table. This summary should also give the detailed results of the isotope pairing incubation (D14 and D15 separately, estimates for anammox and nitrification). A more complete characterization of the sampled sites with an organic matter proxy (TOC, DI, RI, chlorophyll) and ambient concentrations of nutrients would increase the re-usability of the results for other topics.

*We did not measure ambient nutrient concentrations, as these have been commonly reported and are consistently low in this oligotrophic environment.*

*We now mention previously measured concentrations at line 75*

**Previously reported water column nutrient concentrations at this site are $NO_3^-$ 0.05 – 0.7 μM, $NH_4^+$ 0.05 – 1.8 μM, orthophosphate 0.35 – 0.5 μM and sediment organic carbon content is < 0.24% and benthic chlorophyll *a* has been reported as ranging from 11 – 15 mg m$^{-2}$**

Minor comments:
*All comments now addressed.*

- Section 2.1, p. 2, l. 83: Reaction rates are were calculated … -> delete 'are'
- Section 2.2, p. 4, l. 105: course -> coarse?
- Table 1: Are the denitrification rates from SR or FTR?
- Figure 2: Are the error bars of site 1 and 2 missing or just too small to show?
- Figure 6: From which type of experiment are the presented results- SR or FTR?
- Figure 6: Error bars (where applicable) are missing
*Error bars are not available for all studies, so have been omitted for clarity*

**Response to reviewer 2**

In the revised version the authors integrated stirred slurry incubations and come to a different conclusion that now supports the hypothesis of denitrification under bulk oxic conditions for certain types of sediments. The authors rewrote considerable parts of the manuscript and answered all questions thoroughly. I have only a few minor comments:

Line 128: I am still a bit sceptic about the relatively low permeabilities for such coarse sediments. How do the values compare with other values measured in this area? Since the permeabilities are not used for further analysis this is a minor issue, but if you find that literature values of this are usually higher this should be mentioned here.

*We believe these low values are due to the fact that even though the median gran size is higher, there is as small amount of very fine sediment that clogs the sediment and reduces permeability. The values are in the range of $1.6\text{-}6 \times 10^{-11}\ m^{-2}$ previously reported at these sites by Glud, R.N., B.D. Eyre, and N. Patten. (2008). Biogeochemical responses to mass coral spawning at the Great Barrier Reef: Effects on respiration and primary production. Limnology and Oceanography, 53(3): 1014-1024.*

Line 160: the discussion about the steady state condition is confusing. In the FTR, 15Nitrate is present all the time and the oxic conditions change with each step. I would think that 10 min after the O2 at the outlet reaches its plateau the 15N-N2 production should be also in steady state. For the stirred slurry incubations it depends on what you are looking at. The approach itself is based on non-steady state conditions since all the concentrations change over time. However, the exchange between the porous space inside the grains and the porewater should be in quasi steady state. It would be very helpful to rework this part.

*We have now changed this discussion to refer specifically to the FTRs, and not the SRs.*

Spellings etc:
Line 83: "Reaction rates (are?) were….."
Line 162: "..in the stirred reactors , the samples take(n?) were unlikely to represent steady state…"

*All changes made*

[revised manuscript text omitted]